# The Gut Microbiota at Different Developmental Stages of *Apis cerana* Reveals Potential Probiotic Bacteria for Improving Honeybee Health

**DOI:** 10.3390/microorganisms10101938

**Published:** 2022-09-29

**Authors:** Pham Thi Lanh, Bui Thi Thuy Duong, Ha Thi Thu, Nguyen Thi Hoa, Mi Sun Yoo, Yun Sang Cho, Dong Van Quyen

**Affiliations:** 1Institute of Biotechnology, Vietnam Academy of Science and Technology, 18 Hoang Quoc Viet, Cau Giay, Hanoi 11307, Vietnam; 2University of Science and Technology of Hanoi, Vietnam Academy of Science and Technology, 18 Hoang Quoc Viet, Cau Giay, Hanoi 11307, Vietnam; 3Bacterial Disease Division, Animal and Plant Quarantine Agency, 177 Hyeksin 8-ro, Gimcheon-si 39660, Korea

**Keywords:** *Apis cerana*, development stages, gut microbiota, probiotics, 16S rRNA sequence

## Abstract

Honeybees play a vital role in the ecological environment and agricultural economy. Increasing evidence shows that the gut microbiome greatly influences the host’s health. Therefore, a thorough understanding of gut bacteria composition can lead to the development of probiotics specific for each development stage of honeybees. In this study, the gut microbiota at different developmental stages (larvae, pupae, and adults) of the honeybees *Apis cerana* in Hanoi, Vietnam, was assessed by sequencing the V3–V4 region in the 16S rRNA gene using the Illumina Miseq platform. The results indicated that the richness and diversity of the gut microbiota varied over the investigated stages of *A. cenara*. All three bee groups showed relative abundance at both phylum and family levels. In larvae, *Firmicutes* were the most predominant (81.55%); however, they decreased significantly along with the bee development (33.7% in pupae and 10.3% in adults) in favor of *Proteobacteria*. In the gut of adult bees, four of five core bacteria were found, including *Gilliamella apicola* group (34.01%) *Bifidobacterium asteroides* group (10.3%), *Lactobacillus* Firm-4 (2%), and *Lactobacillus* Firm-5 (1%). In contrast, pupae and larvae lacked almost all core bacteria except *G. apicola* (4.13%) in pupae and *Lactobacillus* Firm-5 (4.04%) in larvae. This is the first report on the gut microbiota community at different developmental stages of *A. cerana* in Vietnam and provides potential probiotic species for beekeeping.

## 1. Introduction

Honeybees are of global importance owing to their pollination services; they pollinate hundreds of billions of dollars of crops annually [1,2]. However, the recent decline in bee populations has promoted research on potential agents that affect bee health, including nutrition, toxins, pathogens, parasites, and especially their microbiota, for the sustainable development of the honeybee industry [1,2,3,4,5]. Several studies have shown that bacteria in the bee gut help their host defense against invading pathogens [6,7,8]. Therefore, understanding this microbial community will undoubtedly offer novel insights into the development of probiotics for beekeeping to improve bee health and, more generally, into crucial unresolved aspects of host-microorganism symbiosis [1,9].

Bee gut microbiota plays a critical role in host nutrition, weight gain, endocrine signaling, immune function, pathogen resistance, and even bee social behavior [2]. In contrast, perturbation of the microbiota can lead to negative repercussions for host fitness [2,10,11,12,13,14]. Recently, honeybees have been considered tractable models for gut microbiota research to unravel how gut communities affect their hosts and to understand general principles of the processes that determine gut community composition and dynamics [1,2,15]. In addition, the availability of microbiota-free hosts without chemical surface sterilization or antibiotics enables the investigation of how the microbiota influences host phenotypes, including disease states [2].

Previous studies have shown that the gut microbial community of worker honeybees (*A. melifera* and *A. cerana*) is dominated by 5–9 taxa belonging to four phyla, including *Firmicutes*, *Actinobacteria*, *Proteobacteria*, and *Bacteroidetes* [1,2,12,16,17,18]. Typically, the gut microbiota of the adult honeybee consists of five core bacteria, including the beta proteobacterium *Snodgrassella alvi*, the gamma proteobacterium *Gilliamella apicola*, *Lactobacillus* Firm-4, *Lactobacillus* Firm-5, and the *Bifidobacterium*. In addition, *Firischella perrara*, *Bartonella apis*, *Apibacter adventoris*, and *Parasaccharibacter apium* are sometimes present at variable levels (1–7%) [1,2,5,6,7,17], and smaller numbers of bacteria often represent environmental bacteria [6].

Although many efforts have been made toward the understanding of honeybee gut microbiota, information on the microbial communities present in the gut of larvae and pupae is still conflicting and inconsistent [9]. This study aimed to investigate the variation of gut microbiota in *Apis cerana* larvae, pupae, and adults cultivated in Vietnam using high-throughput 16S rRNA gene sequencing to unravel the function of these bacteria in each developmental stage of their hosts and provide a valuable genetic resource for the development of probiotics to improve bee health.

## 2. Materials and Methods

### 2.1. Sample Collection

The larvae, white pupae, and adults (3 samples per group) of *A. cerana* were collected in Hanoi, Vietnam. The samples were stored at −20 °C for subsequent molecular analysis.

### 2.2. DNA Extraction

The DNA of the gut microbiota was extracted using the DNA extraction method mentioned in our previous study [16]. Briefly, 10 g of each bee sample (larvae or pupae or abdomen of adults) were homogenized in sterile DNA extraction buffer (100 mM Tris HCl, 50 mM EDTA, 50 mM NaCl, 1% SDS, pH 7.0) using a sterile elastic pestle and centrifuged at 2000 rpm in 5 min. The aqueous upper phase was transferred into a new microcentrifuge tube with 50 μL of protease K (20 mg/mL) and 15 μL of lysozyme (100 mg/mL), incubated at 65 °C for an hour, and then centrifuged at 5000 rpm in 5 min. The aqueous upper phase was collected, and phenol/chloroform/isoamyl (25:24:1) was added with a ratio of 1:1 (*v*/*v*), mixed, and centrifuged at 12,000 rpm for 20 min. The aqueous upper phase was transferred into a new microcentrifuge tube. The DNA was precipitated using isopropanol with a 1:1 (*v*/*v*) ratio for 30 min at room temperature centrifuged at 12,000 rpm for 20 min at 4 °C. The DNA pellet was washed twice with 70% ethanol and then dried by Speedvac (Thermo Scientific, Waltham, MA, USA) for 10 min. The DNA was re-suspended in nuclease-free water. The final DNA concentration and purity were determined by a Nanodrop 2000 UV-vis spectrophotometer (Thermo Scientific, Waltham, MA, USA), and DNA quality was checked by 1% agarose gel electrophoresis.

### 2.3. Illumina MiSeq Sequencing

The V3-V4 hypervariable regions of the bacterial 16S rRNA gene were amplified with the primers 341F (5′-TCGTCGGCAGCGTC-AGATGTGTATAAGAGACAG-CCTACGGGNGGCWGCAG-3′) and 805R (5′-GTCTCGTGGGCTCGG-AGATGTGTATAAGAGACAGGACTACHVGGGTATCTAATCC-3′). The thermal profile for the PCR was a cycle at 95 °C for 3 min and 25 cycles of 95 °C for the 30s, followed by 55 °C for 30s and 72 °C for 30s, and a final cycle at 72° for 5 min. Secondary amplification for attaching the Illumina NexTera barcode was performed with i5 forward primer (5′-AATGATACGGCGACCACCGAGATCTACAC-XXXXXXXX-TCGTCGGCAGCGTC-3′; X indicates the barcode region) and i7 reverse primer (5′-CAAGCAGAAGACGGCATACGAGAT-XXXXXXXX-GTCTCGTGGGCTCGG-3′). The condition of secondary amplification is the same as the former, except for the amplification cycle set to 8. The PCR product was confirmed by 1% agarose gel electrophoresis, visualized under a Gel Doc system (BioRad, Hercules, CA, USA), and purified with the CleanPCR (CleanNA, Waddinxveen, The Netherlands). The product quality and size were assessed on a Bioanalyzer 2100 (Agilent, Santa Clara, CA, USA) using a DNA 7500 chip. The sequencing was carried out at Chunlab, Inc. (Seoul, Korea) with Illumina MiSeq Sequencing System (Illumina, San Diego, CA, USA) according to the manufacturer’s instructions.

### 2.4. 16S rRNA Gut Community Analysis

The raw data or paired-end reads generated by Illumina MiSeq were checked the quality (QC) and filtered out the low-quality reads (<Q25) by Trimmomatic v0.32 [19]. The quality-controlled reads were then used for merging paired-end reads using PANDAseq [20], followed by a primer trimming process using ChunLab’s in-house program at a similarity cut-off of 0.8. HMMER’s hmmsearch program was used to detect the 16S rRNA non-specific amplicons [21]. The sequences were then denoised by DUDE-Seq [22]. The non-redundant reads were extracted through UCLUST-clustering [23]. Sequences were then subjected to taxonomic assignment by USEARCH using the EzBioCloud database and more precise pairwise alignment [24]. UCHIME [25] and the non-chimeric 16S rRNA database from EzBioCloud were used to check chimeras. Species-level identification was made based on the cut-off of 97% similarity of 16S rRNA gene sequences. CD-HIT [26] and UCLUST5 were performed for sequence clustering. The alpha diversity indices and rarefaction curves were conducted by in-house code. Relative abundance plots were generated using bacterial phylum and families with a cut-off of 1%.

### 2.5. Statistical Analysis

Statistical analyses were performed and visualized by R software (version 4.0.5). Relative abundance plots were generated using bacterial phylum and families with a cut-off of 1%. The α-diversity of the honeybee gut microbiome was measured using the ACE, Chao, Shannon, Simpson, and Phylogenetic Diversity (PD) indices. One way-ANOVA was performed, followed by Tukey’s HSD test to see if any group differed significantly further from the others for Shannon and Simpson indices (parametric tests), whereas the Kruskal–Wallis test was performed for the others (non-parametric test).

## 3. Results

### 3.1. Summary of NGS-Sequencing

By Illumina MiSeq sequencing analysis, 310,395 valid sequences were obtained with an average sequence length of 422 bp. As a result, a total of 2226 bacterial operational taxonomic units (OTUs) were identified at the 97% sequence similarity cut-off. The Good’s coverage index calculated the OTU% in all samples with an average bacterial coverage was 0.99. The averages of OTUs were highest in adults and lowest in pupae, and a significant difference in the average OTU number among different developmental stages of the honeybees was detected (*p* = 0.03) with strong evidence for the difference between the adult and pupal groups (*p* = 0.0044) (Figure 1).

### 3.2. Gut Bacterial Community Diversity Indices

Diversity indices include species richness (ACE, Chao) and species evenness (Shannon, Simpson), which are the measures of species diversity based on the number and pattern of OTUs found in the sample, and phylogenetic diversity measures the biodiversity with the phylogenetic difference between species by the total of the lengths of all those branches. The results showed sufficient diversity of bacterial composition in the samples (Figure 1). Thus, we assessed the bacterial community in all samples. Moreover, there was a significant difference among three groups in terms of alpha diversity, such as ACE (*p* = 0.03), Chao (*p* = 0.027), Shannon (*p* < 0.001), Simpson (*p* = 0.031), and PD (*p* = 0.03). Among the three groups, the highest bacterial diversity and richness were found in the adult gut, with the highest values in ACE, Chao, Shannon, and PD indices and the lowest value in Simpson index (Figure 1). The results of multiple comparisons showed that the difference between the adult and pupal groups was significant in four diversity indices (all *p* < 0.01), exception of Simpson index. In addition, there were significant differences in the Shannon index by intergroup comparisons with *p_adults-larvae_* = 0.00005, *p_adults-pupae_* = 0.00174 and *p_larvae-pupae_* = 0.00358 (Figure 1).

### 3.3. Microbial Flora at Different Developmental Stages

The remarkable differences in taxon richness and relative abundance patterns through the developmental stages of the honeybee at both phylum and family taxonomic classification levels were analyzed. A total of 4 major phyla and 16 bacterial families were identified. The major phyla included *Proteobacteria*, *Firmicutes*, *Actinobacteria*, and *Bacteroidetes* (Figure 2), whereas the dominant families were *Acetobacteraceae* (*Bomlella* sp., *Commensalibacter* sp.), *Bifidobacteriaceae*, *Enterobacteriaceae*, *Enterococcaceae*, *Lactobacillaceae*, *Neisseriaceae* (*Snodgrassella* sp.), *Orbaceae* (*Gilliamella* sp.), *Rhizobiaceae* (*Bartonell* sp.), *Bacillaceae*, *Lachnospiraceae*, and *Sphingomonadaceae* (Figure 3). All three groups showed the relative abundance in the gut flora composition at both the phylum level and family level and dominant bacteria. *Firmicutes* were the dominant phyla for the larvae, but their relative abundance decreased significantly upon bee development and was largely replaced by *Proteobacteria* (Figure 2). In the pupal and adult stages, an increase in gut flora diversity at the phylum level was detected. The *p*-values were determined by an intergroup comparison and they were as follows: *Proteobacteria* (*p* < 0.001), *Firmicutes* (*p* < 0.001), *Actinobacteria* (*p* < 0.001), and *Bacteroidetes* (*p* < 0.001).

The larval gut microbiota exhibited a low bacterial diversity, which included two main phyla, including *Firmicutes* (81.55%), *Proteobacteria* (17.0%), and other bacteria (Figure 2). Three dominant families were *Lactobacillaceae* (47.9%), *Enterococcaceae* (33.0%), and *Acetobacteraceae* (15%) (Figure 3). *Lactobacillus kunkeei* group (30.11%) and *Melissococcus plutonius* (25.03%) were the most abundant group or species, followed by *Lactobacillus_uc* (13.04%), *Commensalibacter AY370188_s* (8.23%), *Enterococcus faecalis* (8.05%*)* and *Bombella intestinii group* (6.94%) (Figure 4). Remarkably, *M. plutonius* is the causative agent of European Foulbrood (EFB) in honeybees, and *E. faecalis* is a common secondary invader associated with EFB.

In the gut flora of pupae, *Firmicutes* considerably decreased to 58.6% (*p* < 0.001), while *Proteobacteria* significantly increased to 33.7% (*p* < 0.001) in comparison to those in the larval bee gut (Figure 2). Furthermore, the phylum diversity at this stage slightly increases with the relative abundance of phyla *Bacteroidetes* (0.1%) and other bacteria. The most abundant families include *Bacillaceae* (51.8%), *Sphingomonadaceae* (15.3%), *Lachnospiraceae* (8.16%), and *Alcaligenaceae* (5.8%) (Figure 3). Incredibly, three groups of bacteria mainly resided in the pupae gut, which were not found or presented at low percentages in other stages, including *Bacillus subtilis group* (50.1%), *Sphingomanas pruni group* (15.3%), and *Lachnospriraceae_uc* (8.16%) (Figure 4).

The gut microbiota of adult honeybees showed the highest diversity at both the phylum and family levels. Additionally, the number of species substantially increased. It mainly consisted of four phyla *Proteobacteria* (70.3%), *Actinobacteria* (10.4%), *Firmicutes* (10.3%), and *Bacteroidetes* (8.2%) (Figure 2). The relative abundance of bacterial families in worker microflora were *Orbaceae* (43.1%), *Enterobacteriaceae* (13%), *Bifidobacteriaceae* (11%), *Lactobacillaceae* (9.5%), *Bacillaceae* (6.2%), *Rhizobiaceae* (6%), *Neisseriaceae* (6%), and *Porphyromonadaceae* (2%) (Figure 3). In the gut of adult bees, four out of five core bacteria were found, including *G. apicola* group (34.01%), *Bifidobacterium asteroides* group (10.3%), *Lactobacillus* Firm-4 (2%) (*Lactobacillus melis*), and *Lactobacillus* Firm-5 (1%) (*Lactobacillus apis*, *Lactobacillus helsingborgensis*, *Lactobacillus kimbladii*), whereas pupae and larva gut lack almost all core bacteria, except 4.13% of *G. apicola* found in the gut flora in pupae and 4.04% of *Lactobacillus* Firm-5 (*L. kimbladii*) in the larval gut (Figure 4). In addition, the gut microbiota at the adult stage showed colonization of some non-core phylotypes such as *Bartonella apis* group (6.0%), *Apibacter mensalis* (6.34%), and *L. kunkeei* group (5.15%) (Figure 4). Additionally, adult gut microbiota was associated with opportunistic environmental bacteria, including members of the *Enterobacteriaceae* (10%) (5% *Enterobacteriaceae group*, 4% *Enterobacterales_uc*, 1% *Klebsiella* FWNZ_s). These opportunistic bacteria were not present in the larval gut or presented at a low level in the pupal abdomen (0.94% of the *Enterobacteriaceae group*).

By comparing the gut bacterial composition, we found that the *Lactobacillus* spp. was the most abundant but highly variable among the three bee groups (*p* < 0.001). The same trend was observed in the relative abundance of other bacterial groups, including *Bacillus subtilis group* (*p* < 0.001) and *G. apicola group* (*p* < 0.001).

## 4. Discussion

The microbiome is considered one of the most critical factors that shapes the life history of its hosts [2,14]. Detailed information on the overall composition, function, and persistence of the gut microbiota in honeybees is necessary before probiotics can be introduced into the beekeeping practice [8,9]. In other words, knowing which bacteria are suitable for each developmental stage of honeybees can lead to the development of probiotics specific for each stage.

Though new honeybee larvae are devoid of bacteria, they are fed by workers with nectar, pollen, honey, etc., which may lead to an accumulation of bacterial species from hive materials in their blind gut [2]. In this study, we found that the larvae hosted significantly more *Firmicutes* than pupae (*p* < 0.001) and adults (*p* < 0.001). *Firmicutes* were four times more abundant than *Proteobacteria* in the larvae gut. Meanwhile, in the pupal and adult stage, the microbiota favored *Proteobacteria* (*p* < 0.001). It indicates that the relative proportions of the core microbiota in the colonies vary with the age of honeybees found in a recent study [9]. Moreover, our results also reinforced the previous finding that the dominant role of *Proteobacteria* in the insect gut microbiota may be an exclusive feature of insect gut microbiota composition [27].

Further characterization showed that larvae and pupae have a few core bacteria in their gut. This was consistent with previous reports showing that the core bacteria are relatively rare in larvae; however, they consist of highly variable communities dominated by non-core environmental bacteria [6,28]. Indeed, in this study, the microbiota of larvae was predominant by *L. kunkeei*, *M. plutonius*, *Lactobacillus_uc*, *Commensalibacter,* whereas those in pupae were *B. subtilis*, *S. pruni*, *Lachnospiraceae*. All these bacteria can protect the host from pathogens [29,30,31], except for *M. plutonius* [4] and *S. pruni* [32]. For instance, *L. kunkeei*, a well-known fructophilic lactic acid bacterium (FLAB), can approve the larvae viability by decreasing the honeybees infected by various pathogens, such as *Paenibacillus larvae*, *Nosema ceranae*, *Ascosphaera apis*, etc. [29,33] due to antibacterial activity via the production of antimicrobial peptides or protein [29]. It has been found that *L. kunkeei* significantly decreased when compared to the gut microbiota of healthy and SD-affected larvae in SBV-susceptible *A. cerana* [28]. *B. subtilis* was one of the most important probiotic *Bacillus* species for humans and animals since they have a broad range of antioxidant and antimicrobial activities (such as surfactin, bacteriocins, bacteriocin-like inhibitory substances (BLISes), etc.) over numerous pathogenic fungal and bacteria and posed considerably good probiotic features [34,35]. *Lachnospiraceae* had a considerable ability to break-down diet-derived polysaccharides, such as starch, inulin, and arabinoxylan [36].

Interestingly, an endosymbiotic acetic acid bacterium in bumble bees, *Bombella intestine,* was also detected with an adequate amount (7%) in the larvae since this species was first identified in *A. cerana* in Korea by Yun et al. (2022) [28]. These bacteria can increase larval survival against pathogens by the secretion of acids [37]. It has been reported that *B. intestini* and *Lactobacillus_uc* were only detected in sacbrood virus-resistant *A. cerana,* whereas they were not included in SBV-susceptible *A. cerana* larva [28].

Noticeably, *M. plutonius* (*Lactobacillales*, *Enterococcaceae*) is the bacterial agent causing European foulbrood (EFB) in honeybee larvae and entering the intestinal tract of honeybee larvae through contaminated food provided by adult bees [4,6,38]. In this study, we found *M. plutonius* with a relative abundance of 25.03% though larval samples and their colonies did not show any clinical signs. It has been suggested that *M. plutonius* virulence differs in the individual insect level among *M. plutonius* strains with different genetic backgrounds and at both brood frame and colony levels in honeybees [38]. Then, mortality of the honeybee brood caused by the *M. plutonius* isolates varied greatly [4]. Sometimes, it may require secondary invaders to produce symptoms in larvae (*Paenibacillus alvei*, *E. faecalis*, *Brevibacillus laterosporus,* or *Achromobacter eurydice*) [4,38]. Potentially, the high abundance of *Lactobacillus* sp. (*L. kunkeei*) and other environmental bacteria (*Commensalibacter*, *Bombella*) in the larval gut inhibited the colonization of *M. plutonius* and its secondary invaders, as suggested in previous studies [29,31]. Indeed, our results showed that the prevalence of *M. plutonius* and *E. faecalis* in EFB susceptible larvae were significantly higher than those in normal larvae (data not shown).

Adult workers have a relatively diverse set of bacteria in their gut compared with larvae and pupae. The gut microbiota of *A. cerana* adult workers consisted of nine bacterial species, which was consistent with our recent studies on the gut microbiota of Vietnamese *A. cerana* [16] and the previous reports that the gut microflora of workers (*A. melifera* and *A. cerana*) may contain from five to nine taxa belonging to four phyla, including *Firmicutes*, *Actinobacteria*, *Proteobacteria*, and *Bacteroidetes* [1,2,12,16,17,18,28]. However, the composition and the abundance of predominant bacteria groups varied due to geographical differences and natural conditions.

There were four ubiquitous core bacteria found in Vietnamese *A. cerana* adults, including *G. apicola*, two members of *Lactobacillus* (*L.* Firm-4 (*L. melis*) and *L.* Firm-5 (*L. apis*, *L. helsingborgensis*, *L. kimbladii*)), and a *Bifidobacterium* species. *Gilliamella* spp. is endosymbionts and plays a role in degrading polysaccharides that could affect the absorption of host nutrients [28]. *Gilliamella* was the most dominant bacterial in the gut microbiota of *A. cerana* adult workers in Vietnam. Its relative abundance revealed a key role for *Gilliamella* as a core genus in workers [30]. They are endosymbionts and are the most critical types of fermentation bacteria in the honeybee intestine, protecting honeybees from pathogens and improving dietary tolerances [28,39].

*Lactobacillus* is the most critical probiotic genus in the digestive tract of honeybees, with essential roles in carbohydrate metabolism [40] and protecting hosts through producing antimicrobial metabolites (such as organic acids, diacetyl, benzoate, and bacteriocins) and induction of immune responses [2,6,30,39]. Recent studies showed that *L. Firm*-4 and *L. Firm*-5 hosted numerous phosphotransferase systems that participated in the digestion of sugars, particularly in *L.* Firm-5. Moreover, oral supplementation with bee gut *Lactobacillus* upgraded glycerophospholipid levels in the hemolymph and stimulated the memory of bumblebees; specifically, *L*. Firm-5 can enhance learning and memory behaviors in *A. mellifera* by the indole-AhR signaling pathway [40]. The colonization rule of *Lactobacillus* fluctuated with changes in nutrition, hive, and social environment [30,41]. Our results showed that the relative abundance of *Lactobacillus* in the gut microbiota was altered in different development stages of honeybees, *A. cerana.* It suggests that honeybees need a different amount of *Lactobacillus* for their development at each stage.

*Bifidobacterium (B. asteroids*) can stimulate the production of host hormones (juvenile hormone III) known to impact bee development [14,17,40]. The abundance of this bacterial genus varied in workers, and they were considered a bacterial indicator for assessing the age of workers [30]. Bee-associated *B. asteroids* and *L.* Firm-5 have a set of genes essential for biosynthesis and assimilation of trehalose that is used for energy storage in insects [1]. *G. apicola*, *L.* Firm-4, *L.* Firm-5, and bee-associated *B. asteroides* all can utilize glucose and fructose, the significant sugars in the bee diet [1,28]. In addition, a common non-core bee gut species is *B. apis*. This honeybee gut symbiont was diverse in healthy bees close to bees from collapsing colonies suggesting their positive effect against host diseases [6]. In this study, the adequate amounts of two of these bacteria found in the gut of adult honeybees proved their importance in honeybees’ health.

Recent studies indicated that the honeybee has the potential to be a source of new bacteria with probiotic properties for the beekeeping industry [1,8,13,42,43]. For instance, there was a correlation between the presence of *Bifidobacteria* and other lactic acid bacteria strains (*L. plantarum* Lp39, *L. rhamnosus* GR-1, *L. kunkeei* BR-1, *L. kimbladii*, *L. melis*, etc.) and *B. subtilis* Mori2 with the absence of the larval pathogens *M. plutonius* and *P. larvae* [7,44,45,46]. In addition, the mixture of *B. catenulatum*, *B. longum*, *B. asteroids*, *B. indicum*, *Lactobacillus* sp. and *L. kunkeei*; and *L. johnsonii* exhibited in vivo effect against Nosemosis, an important disease caused by *Nosema ceranae* or *Nosema apis* in adult bees [42]. Though limited progress was made, probiotics might be a potential alternative to the common use of chemical treatments or antimicrobial drugs in beekeeping [8,31].

It has been known that gut microbiota plays an important role in nutrition and health. Our data suggested that the gut bacteria changes at different life stages of honeybee *A.cerana* can modulate the honeybee’s development via changing the gut bacteria. This finding was supported by previous studies, which indicated that the ontogenetic stage of the honeybee could be an essential factor affecting changes in the gut microbial community [9]. These changes may be associated with nutrition and metabolism, genetic specificities, and environmental factors. Therefore, the appropriate microbial composition in each stage can accelerate the later development and increase the capacity to defend against potential pathogens by space exclusion or nutrient competition [1]. In addition, it has been indicated that the bee gut microflora has numerous parallels to the human gut microbiome, including specificity and evolutionary adaptation to hosts, transmission through social interactions, strain variation, pathogen resistance, role in fermentation and SCFA production, as well as a history of antibiotic exposure, and these similarities may be helpful for initiating general rules of microbiota function [2]. Therefore, our finding suggested the analog of the human microbiota over the development stages and probiotic development for human use.

## 5. Conclusions

In conclusion, our study figured out the changes in the gut bacterial components at different life stages of honeybees, *A. cerana*. We found a high relative abundance of *L. kunkeei*, *Lactobacillus_uc*, and *B. intestinii* in the larvae; *B. subtilis* and *Lachnospiraceae_uc* in the pupae; and *G. apicola*, *B. asteroids*, and *Lactobacillus *spp. (*L. kunkeii*, *L. melis*, *L. apis*,…) in the adults. These bacteria have been known to play a vital role in maintaining the health of the honeybee population and can be used for the development of probiotics for each developmental stage of the honeybee.

## Figures and Tables

**Figure 1 microorganisms-10-01938-f001:**
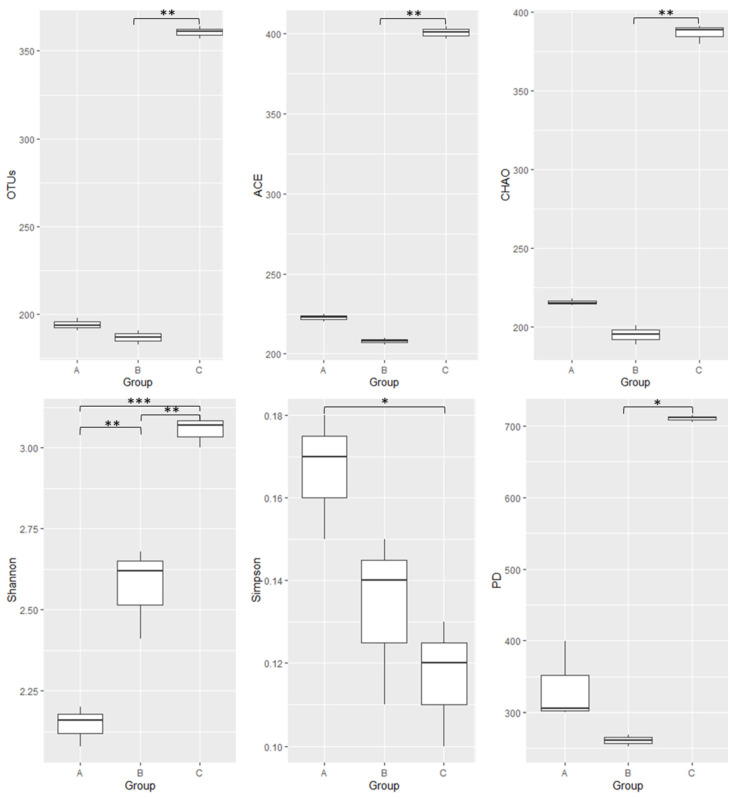
The average OTUs and alpha diversity indices of gut microbiota in larval (A), pupal (B), and adult (C) honeybees, *Apis cerana*. The alpha diversity indices include species richness (ACE, Chao, Jackknife), species evenness (Shannon, Simpson), and Phylogenetic diversity (PD). The significant difference between the two groups was marked as a significant difference (0.01 < *p* ≤ 0.05 is marked as *, 0.001 < *p* ≤ 0.01 is marked as **, and *p* ≤ 0.001 is marked as ***).

**Figure 2 microorganisms-10-01938-f002:**
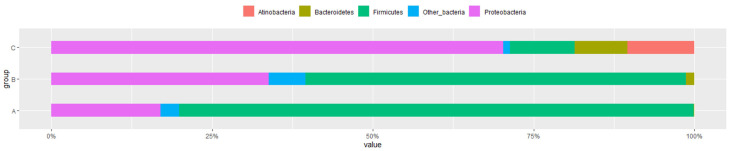
Relative abundance of the major phylum gut microbiota of larval (A), pupal (B), and adult (C) honeybees, *Apis cerana,* with cut-off of 1%. The abundance of different phyla in the sample is shown by the color gradient of the color block.

**Figure 3 microorganisms-10-01938-f003:**
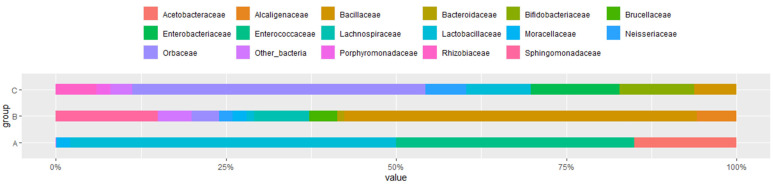
Relative abundance of the major families of gut microbiota of larval (A), pupal (B), and adult (C) honeybees, *Apis cerana,* with cut-off of 1%. The abundance of different families in the sample is shown by the color gradient of the color block.

**Figure 4 microorganisms-10-01938-f004:**
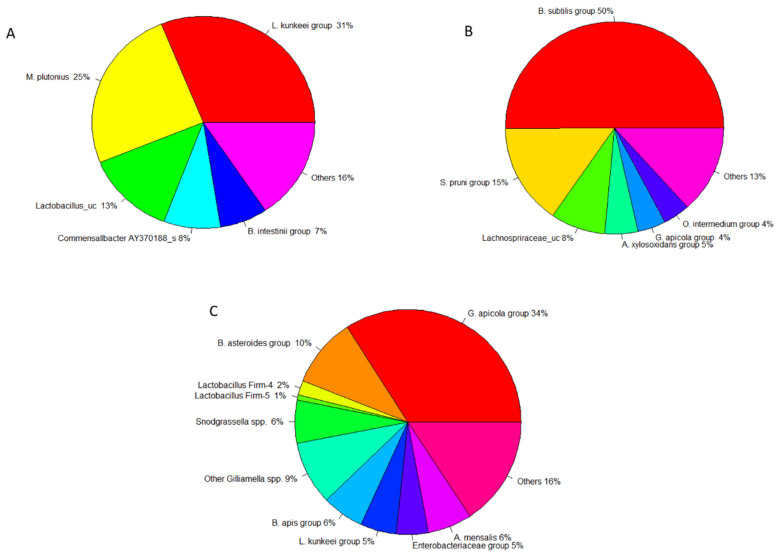
Pie charts of relative abundance of the major bacterial groups in the gut microbiota of honeybees, Apis cerana with a cut-off of 1%: (**A**) larvae, (**B**) pupae, and (**C**) adults.

## Data Availability

The datasets generated during and/or analyzed during the current study are available from the corresponding author upon reasonable request.

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
