# Peer review of "The Gut Microbiota at Different Developmental Stages of Apis cerana Reveals Potential Probiotic Bacteria for Improving Honeybee Health"

_microorganisms, 2022, doi:10.3390/microorganisms10101938_

Round 1
Reviewer 1 Report
I believe this study is meaningful as a study of the gut microbiota at different developmental stages of honey bee.
For future readers, I hope you can describe how this gut microbiota research can be used for human microbiota research.
Reviewer 2 Report
The manuscript presents an interesting study with practical applications about the variation of gut microbiota in Apis cerana at different life stages (larvae, pupae, and adults) which could have results for improving honeybee health.
Comments
Line 178 – Figure 2 - Actinobacteria
ETC - I think that instead of "ETC" the expression "other bacteria" should be used, which would be better to be mentioned at the end, after Proteobacteria, Firmicutes, Actinobacteria, Bacteroidetes.
pg 5 line 185, 186 - for larval gut microbiota "Three dominant families were Lactobacillaceae (47.9%), Enterococcaceae (35%), and Acetobacteraceae (15%) (Fig.3)" - if Firmicutes represent 79.4%, and Lactobacillaceae (47.9%) and Enterococcaceae (35%) together represent 82%, how were these percentages calculated?
pg 6 line 202 – Sphingomanas pruni - Sphingomonas
202 - Sphingomonadaceae is (15%), but Sphingomonas pruni group is (15.3%), greater than 15%, how were these values calculated? Something seems incorrect.
202- Lachnospiraceae (8%), but Lachnospiriraceae_uc (8.16%), - how these values were calculated, is not clear
258, 260 - L. plutonius - M. plutonius
- The conclusions are too broad, the authors should show what are the most significant results obtained
Reviewer 3 Report
This manuscript reported the gut bacteria changes at different life stages of honeybee A.cerana, suggesting an important role of gut bacteria in honeybee’s development. It provides useful information for beekeeping. However, the study did not investigate whether the gut bacteria changes at different life stages of honeybee A.cerana modulate the honeybee’s development via changing the gut bacteria. Thus, it is better to add relative information.
